# Immunoglobulin M Paraproteinaemias

**DOI:** 10.3390/cancers12061688

**Published:** 2020-06-25

**Authors:** Louis-Pierre Girard, Cinnie Yentia Soekojo, Melissa Ooi, Li Mei Poon, Wee-Joo Chng, Sanjay de Mel

**Affiliations:** 1School of Medicine, University of Aberdeen, Aberdeen AB25 2ZN, UK; louispierre.girard.14@abdn.ac.uk; 2Aberdeen Royal Infirmary, NHS Grampian, Scotland AB25 2ZD, UK; 3Department of Haematology-Oncology, National University Cancer Institute, National University Health System, Singapore 119228, Singapore; cinnie_yentia_soekojo@nuhs.edu.sg (C.Y.S.); melissa_ooi@nuhs.edu.sg (M.O.); Michelle_Poon@nuhs.edu.sg (L.M.P.); wee_joo_chng@nuhs.edu.sg (W.-J.C.); 4Cancer Science Institute of Singapore, National University of Singapore, Singapore 117599, Singapore; 5Department of Medicine, Yong Loo Lin School of Medicine, National University of Singapore, Singapore 119077, Singapore

**Keywords:** immunoglobulin M, paraproteinaemia, Waldenstrom macroglobulinaemia, multiple myeloma, lymphoma

## Abstract

Monoclonal paraproteinaemia is an increasingly common reason for referral to haematology services. Paraproteinaemias may be associated with life-threatening haematologic malignancies but can also be an incidental finding requiring only observation. Immunoglobulin M (IgM) paraproteinaemias comprise 15–20% of monoclonal proteins but pose unique clinical challenges. IgM paraproteins are more commonly associated with lymphoplasmacytic lymphoma than multiple myeloma and can occur in a variety of other mature B-cell neoplasms. The large molecular weight of the IgM multimer leads to a spectrum of clinical manifestations more commonly seen with IgM paraproteins than others. The differential diagnosis of B-cell and plasma cell dyscrasias associated with IgM gammopathies can be challenging. Although the discovery of MYD88 L265P and other mutations has shed light on the molecular biology of IgM paraproteinaemias, clinical and histopathologic findings still play a vital role in the diagnostic process. IgM secreting clones are also associated with a number of “monoclonal gammopathy of clinical significance” entities. These disorders pose a novel challenge from both a diagnostic and therapeutic perspective. In this review we provide a clinical overview of IgM paraproteinaemias while discussing the key advances which may affect how we manage these patients in the future.

## 1. Introduction

Monoclonal proteins or paraproteins arise from the clonal expansion of an antibody-secreting B-cell or plasma cell [1]. Plasma cell dyscrasias including monoclonal gammopathy of undetermined significance (MGUS), multiple myeloma (MM), and light chain amyloidosis (ALA) are typically associated with paraproteins [2]. They are also found in mature B-cell neoplasms, most notably Waldenstrom macroglobulinaemia (WM) [3,4]. Paraproteins are routinely identified and characterised using serum protein electrophoresis (SPEP), immunofixation electrophoresis (IFE) and serum free light chain assays (SFLC) [5,6]. These “screening” investigations are often requested during the work up of anaemia, renal impairment, proteinuria, neuropathy and osteoporosis [7]. Identification of a paraprotein based on these investigations typically results in a referral to haematology services for further evaluation. Monoclonal proteins in the absence of symptoms were first described by Dr. Jan Waldenstrom who reported hypergammaglobulinaemia on SPEP of asymptomatic individuals [8]. An increasingly common phenomenon is the detection of paraproteins on health screens when asymptomatic individuals are found to have a raised erythrocyte sedimentation rate (ESR) or globulin fraction and hence undergo screening investigations [7,9].

The majority of referrals for paraproteinaemias are for those of the immunoglobulin G (IgG) or IgA subtypes [7,10]. Though IgM paraproteinaemia only accounts for 15–20% of cases it poses unique diagnostic challenges [7,10]. IgM paraproteins require consideration of a broader range of differential diagnoses as well as unique complications related to the high molecular weight of the IgM pentamer [11]. Hyperviscosity syndrome in patients with WM and immunohaematologic manifestations (discussed in Section 6.6) are notable examples [3,12]. Briefly, large protein molecules such as IgM have high intrinsic viscosity, and even small increments in their serum levels are able to increase plasma viscosity more significantly than IgG or IgA [12]. Hyperviscosity syndrome can also be triggered by type 1 and 2 cryoglobulinaemia, via the same mechanism [12]. Cyroglobulinaemias associated with IgM paraproteinaemias are discussed more comprehensively in Section 6.5.

Peripheral neuropathies are also a common association of IgM gammopathies and are discussed further in Section 6.3 [13]. Figure 1 summarises the recognised clinical manifestations related to IgM paraproteins. In this review, we will provide an overview of the disorders associated with IgM paraproteinaemia and outline our approach to the evaluation of these patients. We will subsequently discuss some of the key advances and challenges in this field.

## 2. Summary of WHO and IMWG (International Myeloma Working Group) Defined Disease Categories Associated with IgM Paraproteins

### 2.1. Immunoglobulin M Monoclonal Gammopathy of Uncertain Significance

IgM MGUS is defined by the International Myeloma Working Group (IMWG) as a serum IgM monoclonal protein of <30 g/L, with a lymphoplasmacytic lymphoid infiltrate in the bone marrow of <10%. Furthermore, there must be no evidence of anaemia, hyperviscosity, lymphadenopathy, hepatosplenomegaly, constitutional symptoms, or other end-organ damage attributable to the underlying lymphoproliferative disorder [14]. IgM MGUS comprises 15–20% all MGUS and in contrast to other subtypes of MGUS is more common in Caucasians than Afro-Caribbean populations [10,15,16]. In a large single-centre study, the median age at diagnosis was 74 years, with a male predominance [16].

Typically, primary progression events include WM, ALA and other B-cell lymphoproliferative disorders (LPD) at a rate of 1.5–2% per year [14,16]. Independent risk factors for progression include the detection of MYD88 L265P mutation and increased levels of serum monoclonal protein [16,17]. Management of IgM MGUS involves clinical monitoring with assessments every 3–6 months including history, physical examination, full blood count, lactate dehydrogenase (LDH), calcium, renal function, IgM and m protein quantification [9]. The frequency of follow-up may be adjusted after 1–2 years depending on the trajectory of the M protein and clinical findings.

### 2.2. Waldenströms Macroglobulinaemia

WM is defined by the histopathologic finding of a lymphoplasmacytic lymphoma (LPL) with an IgM monoclonal protein [18]. WM accounts for over 95% of LPL with non-secretory LPL or LPL with non-IgM paraproteins comprising the remainder [18]. It is a rare disease, with an incidence of 4 per million per year [19]. While immunoglobulin heavy chain translocation or aneuploidies are not described in WM, up to 50% of cases demonstrate a 6q deletion [20]. Mutations in the MYD88, CXCR4, ARID1A, and MLL2 genes have all been described in WM [21]. Among these, the MYD88 L265P mutation is present in close to 90% of cases, the diagnostic significance of which will be discussed in subsequent sections [21].

Symptomatic anaemia, hyperviscosity and immune manifestations are common modes of presentation [19]. Hyperviscosity is more common in WM than MM due to the higher molecular weight of the IgM pentamer compared to IgG or IgA [22]. Paraprotein-related neuropathies can occur in a proportion of patients while organomegaly and lymphadenopathy may be found on physical examination in 15–30% of cases [19]. Immunoparesis also occurs in WM, though is generally milder than in MM [23]. Although recurrent infections have been reported in WM, the degree of immunoparesis does not predict the incidence of infection [24]. More in depth studies of the immune microenvironment in WM are required to further explore the basis for these findings.

Though a variety of rituximab-based regimens (including bendamustine rituximab, cyclophosphamide, dexamethasone, rituximab and bortezomib, dexamethasone rituximab) have shown efficacy against WM, there is currently no consensus that one regimen is superior to others [19,25,26,27,28]. More recently, bruton tyrosine kinase (BTK) inhibition with ibrutinib has shown promising results in both front line and relapsed settings [29,30]. Patients presenting with an IgM paraprotein greater than 30 g/L and a bone marrow LPL infiltrate >10% in the absence of clinical manifestations attributable to the LPL clone are defined as having smouldering WM [19]. The current standard of care for these patients is clinical monitoring [28].

### 2.3. Immunoglobulin M Multiple Myeloma

IgM MM is a rare subtype of MM comprising less than 1% of cases [31]. Clinical features of IgM MM are similar to other subtypes as illustrated in the case above [32]. Although IgM MM may occur as a progression event of IgM MGUS, this is rare [16]. The distinction between IgM MM and WM is important as the treatment for these malignancies differ significantly [32]. The diagnostic criteria for IgM MM are similar to that for other subtypes of MM as defined by the international myeloma working group (IMWG) [14]. The translocation t(11;14) has been reported in close to 40% of patients with IgM MM [32]. Although not specific for IgM MM, this translocation has not been described in WM and hence has clinical utility in distinguishing the two conditions [32]. Gene expression profiling (GEP) has also shown promise as a means for distinguishing these entities and further studies are required to validate this [23]. Treatment for IgM MM is currently similar to that for other subtypes of MM, however given the emergence of venetoclax as a means to target t(11;14), clinical trials evaluating this agent in IgM MM are called for [33]. The prognosis for patients with IgM MM appears similar to that of other subtypes of MM, however this needs to be assessed in prospective studies [32].

### 2.4. Marginal Zone Lymphoma

Marginal zone lymphomas (MZL) are a heterogeneous group of tumours comprising three major subtypes: mucosa-associated lymphoid tissue (MALT) lymphoma, splenic marginal zone lymphoma (SMZL) and nodal marginal zone lymphoma (NMZL) [18]. MZL arise from the malignant transformation of a mature B-cell within the marginal zone of secondary lymphoid follicles, which are most prominent in MALT and splenic lymphoid tissue [34,35]. Though the aforementioned clinical case specifically exemplifies SMZL, IgM paraproteinaemia can also be observed in NMZL and MALT lymphoma [36]. The clinical presentation as well as histopathologic features of MZL can overlap with WM and LPL, and our approach to this diagnostic challenge will be discussed in subsequent sections [37].

### 2.5. Other B-cell and Plasma Cell Dyscrasias Associated with IgM Paraproteins

IgM paraproteins can be associated with a variety of other mature B-cell LPDs [4,38]. Paraproteinaemias have long been recognised in chronic lymphocytic leukaemia (CLL) [39,40]. Recent data suggest that IgM paraproteinaemia in CLL is associated with high risk genetic features and an adverse outcome [41]. Although often not assessed in aggressive lymphomas, approximately 12% of patients with diffuse large B-cell lymphoma (DLBCL) have an IgM paraprotein [42]. It is noteworthy that the finding of IgM monoclonal gammopathy in DLBCL was associated with an adverse prognosis and an increased risk of central nervous system relapse [42,43]. The majority of IgM secreting DLBCL described in a case series by Cox et al. [42] were of the activated B-cell subtype which are known to have an adverse prognosis compared to the germinal centre B-cell subtype of DLBC [42,44]. Although this may partly explain the poor outcome associated with IgM secretion, further studies including GEP and sequencing would be helpful to better understand the biological basis for this finding. Mantle cell lymphoma (MCL) has also been reported to be associated with IgM paraproteins although the clinical implications of this are not known [36]. ALA associated with IgM paraproteins appears to be a distinct entity with a greater incidence of soft tissue and nerve involvement than other subtypes of ALA [45,46]. Although rare, IgM paraproteins are also described in Polyneuropathy, Organomegaly, Endocrinopathy, Monoclonal gammopathy and Skin changes (POEMS) syndrome [47]. It is noteworthy that ALA and POEMS are more commonly associated with IgM lambda compared to IgM Kappa paraproteins [47]. The clinicopathologic characteristics of B-cell and plasma cell disorders associated with IgM paraproteins are summarised in Table 1.

## 3. Approach to the Clinical Evaluation and Initial Investigation of Patients Presenting with IgM Paraproteins

As discussed above, the differential diagnosis for IgM paraproteinaemias is wide and patients may present with a variety of symptoms and signs. Constitutional symptoms such as weight loss, night sweats and fever are suggestive of a lymphoproliferative disorder [51]. Symptoms suggestive of ALA including chronic diarrhoea, early satiety and GI bleeding, along with symptoms of heart failure and postural hypotension, should be elicited [52]. Identifying neuropathy might suggest POEMS or paraproteinaemic neuropathy, whilst headache, vertigo, vision changes and dizziness could indicate hyperviscosity syndrome associated with WM [47,53]. Though less common, if a patient relayed a history of skeletal pain, the possibility of IgM myeloma might come to the fore of the differential diagnosis. The presence of a concomitant urticarial rash might arouse suspicion of Schnitzler syndrome (discussed in detail in subsequent sections) [32,54].

The assessment should also always include a physical examination, in search of lymphadenopathy and hepatosplenomegaly suggestive of a lymphoproliferative disorder. AL amyloid patients might also have macroglossia and periorbital ecchymosis [52]. Although rare, focal skeletal pain may point towards bone lesions occurring in IgM MM [32]. A neurological examination is necessary for the documentation and characterisation of peripheral neuropathy. The presence of neurological symptoms suggesting central nervous system pathology in the context of WM should raise the suspicion of Bing Neel syndrome [55].

Following an appropriate history and examination, basic laboratory tests are warranted. These include a full blood count, peripheral blood film (looking for circulating atypical lymphoid cells), urea and electrolytes, liver function tests, serum calcium and lactate dehydrogenase.

## 4. Overview of Laboratory Techniques Used to Diagnose IgM Paraproteinaemias

### 4.1. Serum Protein Electrophoresis, Immunofixation Electrophoresis and Serum Free Light Chain Assays

SPEP, IFE and SFLC are the standard screening investigations required for the diagnosis of paraproteins [5,6]. During SPEP, serum is applied to a buffered agarose gel matrix across which an electrical charge is applied [56]. This separates albumin from globulin, and further sifts globulin into four major groups: alpha-1-, alpha-2-, beta- and gamma-globulin. Immunoglobulins (with the exception of IgA, which runs in the beta zone) are found in the gamma zone [5,56]. A dense band in the gamma zone is typically seen in monoclonal gammopathies [56]. Though a spike on SPEP is an indication of a monoclonal protein, confirmation of clonality and the isotype of the paraprotein requires IFE [57]. During IFE, serum is prepared as in SPEP and is then incubated with antibodies specific to immunoglobulin light and heavy chains [57]. A monoclonal protein usually results in a discrete band at a specific antibody lane allowing identification of the subtype of paraprotein. SPEP is positive in 87.6% of patients with MM and 73.8% of patients with ALA. SPEP and IFE together are positive in 94.4% and 73.8% of MM and ALA respectively [5]. Figure 2 represents the typical findings on SPEP and IFE indicative of an IgM paraprotein.

SFLC assays target the epitopes on immunoglobulin kappa and lambda light chains which are usually obscured by the normal immunoglobulin conformation [6]. A ratio between labelled kappa and lambda light chains is then calculated. A predominance of one light chain over the other is characteristic of paraproteinaemias [6]. The addition of SFLC to SPEP and IFE significantly improves the sensitivity for paraprotein detection [5,6]. Given the excellent combined sensitivity and specificity of SPEP, IFE and SFLC, we do not routinely recommend that these assays are repeated in the urine.

### 4.2. “M Protein” Quantification

Quantification of monoclonal proteins can be achieved using densitometry of the spike on SPEP [48]. The gel is passed through the densitometer, which optically analyses the density of the gel using the transmission of fixed frequency light. The data collected from the scattering of light can be used with computer software to create an electropherogram, which visually represents the various densities as peaks [1]. Although nephelometry can also be used to quantify immunoglobulins, this assay is not specific for paraproteins and will quantify both the clonal and polyclonal immunoglobulin as a whole [49].

### 4.3. Bone Marrow Studies

Examination of the bone marrow (BM) is often key to the diagnosis of B and plasma cell neoplasms. Morphologic analysis of the BM aspirate may reveal a lymphoplasmacytic lymphoid infiltrate in WM and IgM MGUS while less commonly, a plasma cell infiltrate may be seen, indicating IgM MM [18]. It is noteworthy that a clonal plasma cell infiltrate is also often seen in WM concurrently with the LPL infiltrate [18]. As illustrated in our case of a patient with SMZL, other small B-cell lymphomas can also be identified in this manner. The BM trephine biopsy also provides crucial information on the nature and location of the infiltrate as well as identification of the lineage and clonality of the neoplastic cells by immunohistochemistry [37]. BM examination might be deferred in elderly patients with no symptoms and a monoclonal protein concentration of <15 g/L. In these cases, effective communication with the patient and close clinical monitoring are required [9,50].

### 4.4. Flow Cytometry

Flow cytometric immunophenotyping (FC) is playing an increasingly important role in the diagnosis of haematological malignancies [58]. FC analysis of the peripheral blood is often adequate for the diagnosis of CLL, while FC of BM provides an important adjunct to morphologic assessment of other B-cell lymphoproliferative disorders [18]. FC provides information on the lineage and maturity of the lymphoid cells as well as confirming clonality by surface light chain restriction. FC can also be used to distinguish plasma cells (which would be seen in MM) from small B-cells which would occur in WM or other small B-LPD [58]. Notably, the clonal plasma cells detected in WM usually do not show the aberrant phenotype seen in MM [59]. FC at present cannot reliably distinguish LPL from MZL [60].

### 4.5. Karyotyping and Fluorescent In Situ Hybridisation

Conventional karyotyping and fluorescent in situ hybridisation looking for MM specific translocations should also be performed on BM specimens. The identification of specific chromosomal abnormalities such as t(11;14) will indicate a diagnosis of IgM MM and would not be in keeping with WM [32]. It is noteworthy however that t(11;14) is present in close to 50% of patients with ALA, hence correlation with the overall clinicopathologic picture remains crucial [61]. Other cytogenetic abnormalities, including chromosome 6q deletion and t(9;14), are indicative of WM/LPL, although not specific [20,62].

### 4.6. AL Amyloidosis-Specific Investigations

Should clinical suspicion of ALA with cardiac involvement be aroused, an electrocardiogram, serum troponin, NT-proBNP (N-Terminal pro Brain natriuretic peptide) and an echocardiogram are appropriate [63]. Although biopsy of the involved organ is a consideration, this is often risky, especially in the case of cardiac amyloid, and bleeding complications are greater when renal biopsies are performed in patients with renal amyloidosis [64]. Fat pad aspirates may therefore be a safer initial option to obtain histologic confirmation of amyloid [52]. In the event that the amyloid cannot be subtyped by immunohistochemistry, mass spectrometry based amyloid subtyping may be required [52].

## 5. Utility of Imaging in the Evaluation of Patients with IgM Paraproteins

Computed tomography (CT) and X-ray skeletal survey (SKS) are both recommended modalities of screening for bone lesions in patients with MGUS [65]. The application of these techniques in IgM gammopathies is less clear cut as the progression events often do not involve bone lesions, and the IMWG does not recommend routine bone imaging in the evaluation of IgM MGUS [65]. Given that transformation to IgM MM is however possible (although rare), it is acceptable to use SKS as a screening modality for patients with IgM paraproteins >15g/L or when the free light chain ratio is abnormal even in the absence of symptoms [9]. It is reasonable to defer routine bone imaging with close clinical monitoring in the absence of these risk factors for progression [9]. In patients who have symptoms suggestive of bone lesions and have a normal or equivocal SKS or whole-body CT, whole body magnetic resonance imaging (MRI) may be considered [65]. The role of positron emission tomography (PET) imaging in the evaluation of monoclonal gammopathies remains uncertain, especially when patients have measurable paraprotein and evidence of bone lesions detectable by the imaging modalities discussed above [65]. PET imaging may be considered where extensive extra medullary disease is suspected and in the context of non-secretory plasmacytomas [65].

From the point of view of excluding a lymphoproliferative disorder, a whole-body CT scan with contrast would be considered the modality of first choice with the objective of excluding significant lymphadenopathy or hepatosplenomegaly [66]. Again, deferring this investigation with close clinical follow up in asymptomatic patients is an option. PET imaging would not routinely be required in the absence of clinical features to suggest aggressive transformation of an indolent lymphoma [66]. In these cases, a biopsy of the lymph node concerned would be recommended to confirm transformation. Although serum amyloid P (SAP) scintigraphy is a useful technique for the imaging of amyloid deposits, it is limited by availability [67]. A suggested diagnostic algorithm for IgM paraproteinaemias is presented in Figure 3.

## 6. Challenges Associated with IgM Paraproteinaemias

### 6.1. The Distinction Between LPL/WM and MZL

LPL/WM and MZL are both small B-cell LPDs arising from a post germinal centre B-cell [18]. Whilst WM is defined by IgM paraproteinaemia and a lymphoplasmacytic infiltrate by histopathology, MZL is defined by histological features alone, though it can also be associated with an IgM paraprotein [4,18]. There can be significant overlap in the histopathologic appearance of both disorders, in particular due to plasmacytic differentiation being common in MZL [37,68]. Furthermore, immunophenotyping via immunohistochemistry and FC can show a similar CD5−/CD10− phenotype [37,60]. A paratrabecular pattern of infiltration, presence of LPL cells, Dutcher bodies and mast cells have been proposed to favour LPL over MZL [37]. None of these features are specific however, and the final diagnosis may often remain uncertain.

Discerning between WM and MZL is of importance with regards to prognostication, as MZL and WM have distinct prognostic scores [69,70]. The distinction also has therapeutic significance: the bendamustine/rituximab regimen, for instance, displays efficacy against both MZL and WM, whilst bortezomib based regimens are more efficacious for WM alone [26,27]. Ibrutinib has shown efficacy against both entities, and is approved as front line therapy for WM, but only for relapsed or refractory MZL [29,30,71].

### 6.2. Clinical Utility of MYD88 L265P Mutation Screening

The last decade has seen major advances in our understanding of the molecular biology of indolent B-cell LPDs. One of the key discoveries in this respect has been the identification of the MYD88 L265P mutation in 80–90% of patients with WM, as well as in other B-cell LPDs [21,72]. MYD88 functions as an adaptor protein which activates the NF-κB (nuclear factor κB) and mitogen-activated protein kinase (MAPK) pathways, promoting cell survival [73]. In the context of indolent lymphoid malignancies, the MYD88 L265P mutation is speculated to occur in a post germinal centre B-cell, before or during plasmacytic differentiation [72].

The association between the MYD88 L265P mutation and WM was first made by Treon et al. [21] in 2012, where the mutation was identified by whole genome sequencing in 91% of patients with WM [21]. Interestingly, the frequency of this mutation was lower in non-IgM LPL (25%) indicating that WM is genetically distinct from these less common subtypes of LPL [74]. The frequency of MYD88 mutations in other indolent B-cell malignancies is much lower than in WM, and while 6–13% of patients with MZL have the mutation, it is not found in CLL or MM [75,76]. It has been suggested that these findings could have clinical application in differentiating LPL/WM from other B-cell disorders with similar clinical manifestations [21]. Although MYD88 mutational status can be a useful adjunct to histopathology and flow cytometry, it is not specific for WM, and therefore cannot be applied to reliably distinguish WM and MZL. Although it may be useful to distinguish WM from CLL and IgM MM, this distinction is usually straightforward based on other clinicopathological characteristics.

### 6.3. Neuropathies Associated with IgM Paraproteins

The phenomenon of paraproteinaemic neuropathy occurring in the absence of an active B-cell or plasma cell malignancy has been recognised for many years [13]. More recently, this entity has been categorised under the umbrella of monoclonal gammopathy of clinical significance (MGCS) [77]. There is evidence that IgM paraproteinaemic neuropathy occurs as a result of monoclonal IgM binding directly to neural antigens [78]. Such antigens include myelin-associated glycoprotein, ganglioside, and ganglio-N-tetraosylceramide proteins [78,79,80]. Monoclonal IgM deposits have been detected in myelin fibres and other neurological debris identified within Schwann cells and macrophages, supporting this hypothesis [79,81]. The correlation between antibody titres and the manifestation and resolution of symptoms remains unclear and requires more study [80,82].

IgM-related neuropathy usually presents a picture of progressive and symmetrical distal neuropathy, which can include impairment of both sensory and motor functions [83]. Findings on nerve conduction studies resemble a process of demyelination, and nerve biopsies may reveal axonal loss [13,83]. Screening of patients with “idiopathic” peripheral neuropathy have revealed that 10% have a detectable monoclonal protein [13]. It is particularly important to exclude plasma cell dyscrasias known to cause neuropathy such as ALA and POEMS syndrome, while non haematologic causes of neuropathy should not be overlooked [83]. Distinguishing between paraprotein-related neuropathy and neuropathy related to other medical causes is challenging both based on clinical assessment and nerve conduction studies [83].

The second challenge in paraprotein-associated peripheral neuropathy is the lack of effective management strategies. This is particularly true in the case of IgM MGUS where there is no haematologic indication for clone-directed therapy. Intravenous immunoglobulin might be considered, though there is only evidence of limited, short-term benefit, and the mechanism of action is not well understood [84]. Rituximab may have a role in selected patients, presumably by targeting the underlying B-cell clone [85,86]. Current evidence does not suggest any role for plasmapheresis in the treatment of IgM paraprotein-related neuropathy [83,87]. A single case series has suggested the potential benefit of fludarabine, however given the toxicity of this drug its use may not be justified in this context [88].

### 6.4. Evaluation of the Patient with IgM Paraproteinaemia and Renal Impairment

As discussed above, patients with renal impairment are commonly screened for monoclonal proteins. The challenge in the setting of a patient with a paraprotein and renal impairment is the decision as to whether the renal injury can be attributed to the paraprotein or whether it is related to other medical conditions. As renal impairment is a defining feature of MM and is associated with other monoclonal gammopathies, excluding an active plasma cell or B-cell LPD is crucial [14,89]. Paraprotein-related renal injury can however manifest in the absence of an active B-cell malignancy, a phenomenon defined as monoclonal gammopathy of renal significance (MGRS) [89].

The first step in diagnosing MGRS is to differentiate it from kidney disease unrelated to paraproteinaemia [89,90]. A renal biopsy is therefore recommended in these cases as this distinction can be very difficult based on clinical assessment alone [89,91]. The only subtypes of MGRS described to have IgM deposits in the kidney are type II cryoglobulinaemic vasculitis and, less commonly, proliferative glomerulonephritis (GN) with monoclonal immunoglobulin deposits [89]. Other MGRS lesions including light chain amyloidosis, type 1 cryoglobulinaemic glomerulonephritis, crystal storing histiocytosis and immunotactoid GN can also arise in the context of an IgM paraprotein [89,91].

Achieving an accurate diagnosis requires close collaboration between haematologists, nephrologists and pathologists. The histopathologic diagnosis of MGRS requires expertise and electron microscopy is recommended (although not mandated), making it challenging outside specialised centres [89]. Furthermore, many patients with IgM paraproteinaemias and renal impairment are elderly, and the risk of a renal biopsy may outweigh the benefit, especially in the context of concomitant diseases which may explain the renal impairment. Careful clinical assessment as well as meticulous follow-up and communication with the patient is required in these situations. Management of MGRS depends on the underlying renal lesion and the paraprotein-secreting clone [91]. Although it is accepted that the treatment should be clone directed, there is no high-quality evidence to guide physicians on the appropriate agent or combinations. Regimens used against MM and other indolent B-cell LPDs have hence been extrapolated for the treatment of MGRS [91]. Caution should be exercised especially in the context of IgM paraprotein-related MGRS where the neoplastic clone may be a lymphoplasmacytic lymphoid infiltrate and treatment regimens designed for MM may not be appropriate. It may also be argued that less intensive therapy is required in this setting compared to that used in active MM or WM as the clonal burden is smaller. Further studies are required to better understand the biology of these disorders and determine their optimal treatment.

### 6.5. Autoimmune Manifestations of IgM Paraproteinaemias

Cold agglutinin disease (CAD) comprises 15% of haemolytic anaemias and is characterized by IgM binding to red blood cell (RBC) membranes where it results in complement activation and haemolysis [92]. The large molecular weight of IgM allows it to bind RBCs and cause agglutination more readily than IgG antibodies resulting in spontaneous in vitro agglutination [92]. Although CAD can be associated with infections, it is also a recognized manifestation of IgM gammopathies [92,93,94]. Treatment of patients with CAD and active haemolysis is challenging, given the poor efficacy of most treatment options [92]. Rituximab is a consideration in these situations but only yields a response rate of 40–60% [95]. Cold temperatures can also precipitate cryoglobulinaemia in patients with IgM paraproteins [96]. Cryoglobulins are immunoglobulins which precipitate at low temperatures resulting in endothelial damage and vasculitis of small- to medium-sized vessels [97]. Clinical manifestations include acrocyanosis, Raynaud disease, urticaria, peripheral neuropathy or renal failure [97]. As with many of the other complications of IgM paraproteins, the management of cryoglobulinaemia is based on limited evidence with treatment directed at the underlying disorder [96,97]. Other immuno-haematological phenomena associated with IgM paraproteins include immune thrombocytopenic purpura and less commonly, acquired von Willebrand disease [98,99].

### 6.6. Evaluation of the Patient with IgM Paraproteinaemia and Cutaneous Lesions

A variety of dermatological disorders may be associated with paraproteinaemias and these range from relatively benign conditions, such as xanthoderma, to more destructive lesions such as pyoderma gangrenosum [100,101]. Patients presenting to dermatologists with these dermatoses may be screened for paraproteins and referred for a haematology opinion [101]. The occurrence of skin lesions in association with a paraproteinaemia in the absence of an active B-cell or plasma cell neoplasm has been termed monoclonal gammopathy of cutaneous significance (MGCS) [100]. Skin manifestations of paraproteinaemia can arise via several mechanisms. Primary cutaneous protein deposition can be of intact or modified proteins. Intact proteins may be deposited as entire immunoglobulin molecules (as in macroglobulinosis), or as either light-chain or heavy-chain deposition diseases of the skin [102]. Modified proteins may occur as β-pleated sheets (as in amyloidosis), cryoprecipitated immunoglobulin (as in cryoglobulinaemic vasculopathy), or crystallised deposits (as in crystal storing histiocytosis) [102].

Type 1 cryoglobulinaemia can cause a number of cutaneous manifestations including livedo reticularis and urticaria [103]. Type 2 mixed cryoglobulinaemia is an immune-complex mediated sequelae of IgM paraproteinaemia, associated with vasculitis and skin ulcers [97]. The clonal disorders underlying type 1 and 2 cyroglobulinaemia are not exclusively IgM paraproteinaemias, and they are also associated with other gammopathies and infective causes in the case of the latter [97]. Dermatoses specifically associated with IgM paraproteins are less common. Macroglobulinosis is a rare disorder characterised by skin-coloured papules on the extensor surfaces of the limbs [104]. Cutaneous IgM deposition is the hallmark of this disease which is closely associated with WM [104]. Schnitzler syndrome is characterised by an urticarial rash with a neutrophilic urticarial dermatosis seen on histopathology, in the presence of an IgM paraprotein [105]. Extracutaneous manifestations include hepatosplenomegaly, lymphadenopathy and bone pain secondary to osteosclerotic lesions [105]. Elevated serum interleukin 6 (IL-6) and IL-8 as well as clinical responses to IL-1 antagonists suggests IL signalling is key to the pathogenesis of this disease [106]. Further studies are required to better understand the interplay between the IgM paraprotein and inflamassome activation in this unusual disorder.

As is the case for MGRS and paraproteinaemic neuropathy, the diagnosis of skin lesions related to IgM paraproteins requires a multidisciplinary approach. Given the rarity of these conditions, a high index of suspicion is necessary to avoid overlooking these entities. As in other subtypes of monoclonal gammopathy of clinical significance, the optimal treatment is uncertain. Indeed, a watch and wait approach is appropriate if the skin manifestation is indolent [77]. With regard to targeting IgM-secreting clones, anti-CD20 monoclonal antibody therapy will often be appropriate in the case of a lymphoplasmacytic clone [77]. Bortezomib based regimens may be considered in the case of plasma cell driven MGCS [77]. As has been demonstrated in Schnitzler syndrome, targeting the pro-inflammatory milieu may be an alternative to clone directed therapy [106]. A better understanding of the biology of other MGCS entities may provide avenues for the design of targeted therapeutics specifically for these disorders.

## 7. Future Directions

### 7.1. Genomic Categorisation of WM

Treon et al. [107] highlighted that somatic mutations of the MYD88 and CXCR4 genes were associated with altered clinical presentation and survival in patients with WM [107]. Patients who were wild type (WT) for both genes had an inferior survival compared to MYD88 L265P patients regardless of CXCR4 mutation status. Interestingly, the “double WT” group had lower levels of BM involvement and lower IgM levels compared to the others [107]. MYD88 WT status has also been reported as an independent risk factor for large cell transformation of WM [108]. However, a large retrospective study of WM patients treated mainly with rituximab-alkylator based regimens showed that MYD88 mutational status had no significant impact on long term survival [109]. Therefore, prospective studies are required to clarify the prognostic role of these genetic alterations. The WM IPSS (international prognostic score) is currently used as a prognostic score for WM, although the WM IPSS typically does not change the choice of initial treatment [28,70]. Mutational status in conjunction with clinical parameters may be an attractive means to create a novel prognostic score for WM.

Such a prognostic score could also have therapeutic implications. The differential effect of ibrutinib on the molecular subtypes of WM has been striking: 100% of MYD88 L265P, CXCR4 WT patients responded to ibrutinib, compared to 85.7% and 60% of patients in the double mutant and double WT groups respectively [110]. In a retrospective analysis of WM patients treated with ibrutinib, Abeykoon et al. [111] reported on five MYD88 WT patients, among whom two achieved a partial response, one achieved a minor response, and the other did not respond [111]. These data indicate the potential for use of MYD88 mutation status as a biomarker for response to BTK inhibition. Conversely, in a randomized trial of WM patients comparing ibrutinib-rituximab with single agent rituximab, response rates and survival appeared similar regardless of MYD88 mutational status30. It is noteworthy however that this study was not powered to address the differential efficacy of BTK inhibition across genetic subtypes of WM. Nevertheless, this data suggests that combination with rituximab may overcome the relative resistance to BTK inhibition seen in MYD88 WT WM. Further in vitro and clinical studies are required to address this question. Bortezomib in turn has shown greater efficacy against CXCR4 mutant compared to CXCR4 WT WM [112]. Clinical trials exploring treatment guided by genomic characteristics are called for to validate these findings and provide the evidence base for personalised therapy of WM. Novel therapeutic approaches targeting these mutations are also under investigation and their development is eagerly awaited [113]. Interestingly, WM patients who achieved negativity for peripheral blood MYD88 L265P by allele-specific polymerase chain reaction (PCR) were found to have lower IgM and higher haemoglobin levels [114]. These findings suggest the potential for MYD88 mutational detection as a tool for minimal residual disease assessment.

### 7.2. Novel Techniques to Identify and Quantify Paraproteins

The accurate quantification of IgM paraproteins is of critical importance for diagnosis and response assessment in IgM related B-cell and plasma cell disorders [28,115]. Paraprotein quantification by densitometry of the band on SPEP can yield inaccurate results due to failed protein migration, low protein concentrations, or the presence of cryoglobulinaemic properties [116,117]. Similarly, nephelometric analysis of total IgM may not be accurate at very high IgM concentrations due to non-linearity caused by self-aggregation of the IgM pentamers [117]. Novel techniques to more accurately quantify and monitor paraproteins are therefore required.

The use of mass spectrometry (MS) to quantify monoclonal paraproteins has been proposed based on the fact that paraproteins have a unique molecular mass which allows MS to distinguish them from the polyclonal background [118]. The unique amino acid sequence in the constant region of each heavy and light chain isotype allows top down MS to accurately isotype paraproteins [119]. This technique is currently being applied in the quality control of therapeutic monoclonal antibodies but has not been routinely applied in the clinical setting [120]. Indeed, the clinical applications of MS already include the assessment of protein-based disorders such as haemoglobinopathies and amyloidosis [121,122]. Barnidge and co-workers [119] applied their MS platform liquid chromatography electrospray ionization quadrupole time-of-flight mass spectrometry (LC-ESI-Q-TOF MS) in a case of MM and demonstrated accurate monitoring of disease at levels undetectable by conventional techniques [119]. The “Mass Fix” MS platform has recently been used to assess residual disease post stem cell transplant in MM [123]. Patients who had residual paraprotein by Mass Fix were shown to have an adverse outcome despite being negative for minimal residual disease by next generation flow cytometry. These data suggest that MS has great potential as a clinical tool in the diagnosis and monitoring of paraproteins. Validation of this technique in larger clinical cohorts is called for.

Heavy/light chain immunoassays (hevylite/HLC) identify junctional epitopes that span the heavy and light chain constant domains [117,124]. HLC therefore has the unique property of being able to quantify light chain subtypes of each class of immunoglobulin e.g., IgMκ and IgMλ [124]. Boyle et al. [116] demonstrated that SPEP was only able to identify monoclonal IgM in 85% of WM cases, while the paraprotein was accurately identified by HLC in all the cases analysed [116]. These findings are consistent with studies applying HLC to non-IgM paraproteinaemias, indicating the potential for this technique to be widely applied in dysproteinaemias [125,126].

## 8. Conclusions

Great strides have been made in our understanding of IgM paraproteinaemias in the last decade. This fascinating group of disorders nevertheless continue to pose a diagnostic and therapeutic challenge due to the overlap between clinical entities associated with IgM clones. Close collaboration between haematologists, histopathologists and related medical specialists is vital in arriving at an accurate diagnosis. The importance of correlation between clinical presentation and laboratory findings cannot be emphasized enough in the evaluation of these patients. Advances in molecular diagnostics are bringing us closer to a more refined classification of these disorders with the eventual aim of developing more effective targeted therapeutics to improve patient outcomes.

## Figures and Tables

**Figure 1 cancers-12-01688-f001:**
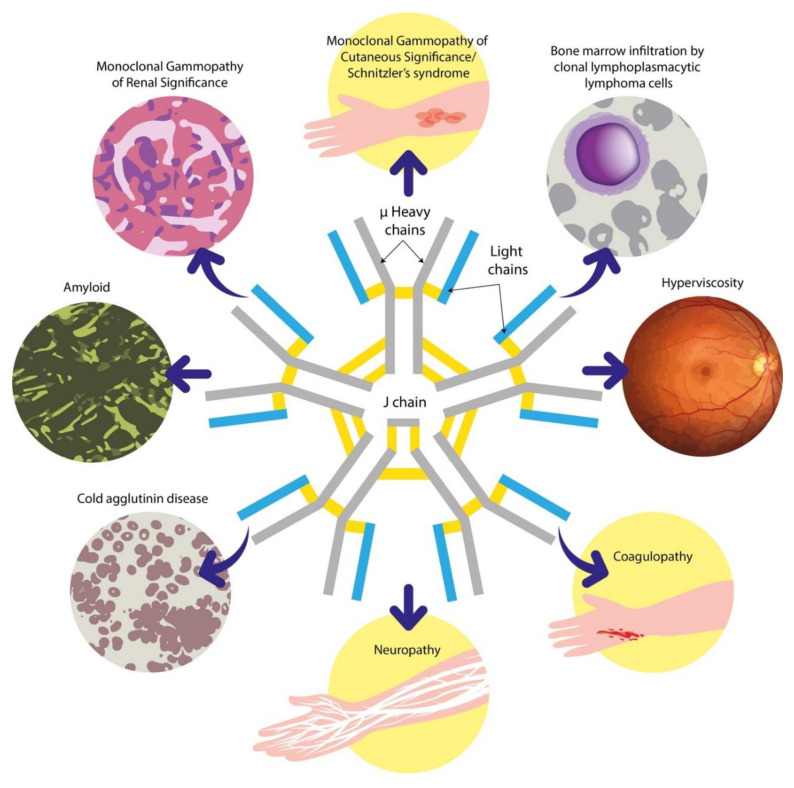
An overview of the clinical manifestations associated with IgM gammopathies. The high molecular weight of the IgM pentamer depicted at the centre is key to the unique behaviour of this paraprotein. IgM = immunoglobulin M.

**Figure 2 cancers-12-01688-f002:**
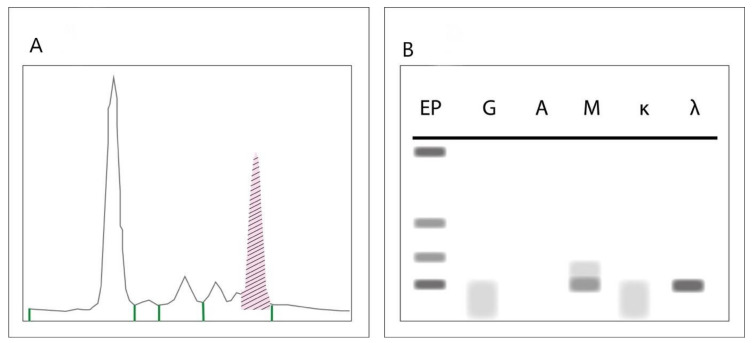
Typical findings on serum electrophoresis (SPEP) (**A**) and immunofixation electrophoresis (IFE) (**B**) indicative of an IgM paraprotein. The shaded area under the SPEP highlights the “spike” characteristic of monoclonal proteins. The discrete bands on IFE confirms clonality and the isotype as IgM lambda.

**Figure 3 cancers-12-01688-f003:**
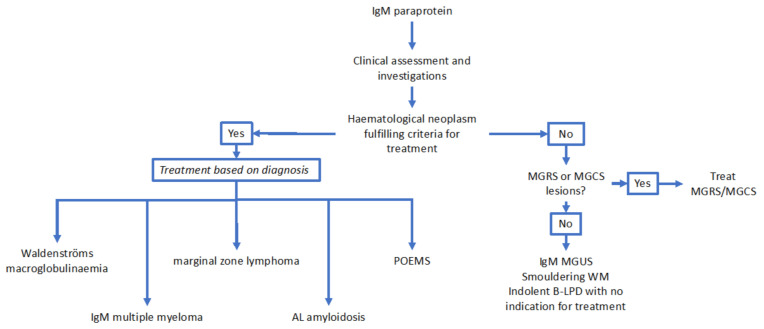
Suggested diagnostic algorithm for IgM paraproteinaemias. Abbreviations: IgM: immunoglobulin M; MGRS: monoclonal gammopathy of renal significance; MGCS: monoclonal gammopathy of clinical significance; MGUS: monoclonal gammopathy of uncertain significance; WM: Waldenströms macroglobulinaemia; B-LPD: B-cell lymphoproliferative disorder; POEMS: polyneuropathy, organomegaly, endocrinopathy, M-protein, skin changes; AL amyloidosis: light chain amyloidosis.

**Table 1 cancers-12-01688-t001:** Clinicopathologic characteristics of haematologic neoplasms associated with immunoglobulin M paraproteins.

Diagnosis	Clinical Presentation	Median Serum IgM Level (mg/dL)	Morphology	Immunophenotype	Cytogenetics	MYD88 L265P Mutation
Waldenströms macroglobulinaemia	Hyperviscosity syndrome Symptomatic anaemia	3215	Lymphoplasmacytic lymphoma/plasma cell	LPL: CD5−/CD10−/CD20+ PC: CD56−	6q deletion in 40%	80–100%
IgM MGUS	Asymptomatic	840	Lymphoplasmacytic lymphoma/plasma cell	LPL phenotype as in WM	Unknown	10–40%
IgM myeloma	CRAB symptoms	4660	Plasma cell	PC: CD38++/CD138+/CD56+	t(11;14) in 40%	0%
Marginal zone lymphoma	Varies by subtype	285	Marginal zone B-cell	CD5−/CD10−/CD20+	t(11;18) in MALT	6–13%
Chronic lymphocytic leukaemia	Often asymptomatic Lymphadenopathy Splenomegaly Anaemia Thrombocytopenia Autoimmunity	400	Small mature lymphocytes	CD5+/CD10−/CD23+	11q, 13q abnormalities, 17p deletion	0%
Light Chain Amyloidosis	Nephrotic syndrome, cardiac failure	800	Plasma cells or mature B-cells	Clonal PC may have a phenotype similar to MM	t(11;14) in 50%	0%

Abbreviations: MGUS: monoclonal gammopathy of undetermined significance; LPL: lymphoplasmacytic lymphoma cells; PC: plasma cell. CRAB: hypercalcaemia, renal impairment, anaemia, bone lesions. MM: multiple myeloma. MALT: mucosa-associated lymphoid tissue. References: [23,24,32,38,40,45,48,49,50].

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
