# Peer review of "Immunoglobulin M Paraproteinaemias"

_cancers, 2020, doi:10.3390/cancers12061688_

Round 1

Reviewer 1 Report

Girard and de Mel et al has done a phenomenal job in reviewing the literature and putting forward a great review of M paraproteinemias.  It is comprehensive and covers the filed well. Literature review is complete. I have following comments which will enhance the quality of this work even more.

  1. In section 7.1, please update the survival data of MYD88L265P in WM. The paper by Treon et al suggested an inferior survival in WM patients with MYD88WT compared to L265P genotype but most of these patients were treated with BTK inhibitor (ibrutinib). However, in patients who were not treated with ibrutinib, the MYD88WT did not suggest poor prognosis. Hence, it is important for providers and for patients to know this message. Abeykoon, AJH 2017. https://pubmed.ncbi.nlm.nih.gov/29080258/.
  2. In section 7.1, please comment on the following matter: there are some instances where a major response was seen with ibrutinib (minority of patients, but now we cannot say that MYD88WT will NOT achieve a major response due to this data: https://onlinelibrary.wiley.com/doi/full/10.1111/bjh.16168). Although this study is a retrospective study, it is important to get this message in this review.  Moreover, when rituximab is added to ibrutinib, the treatment response which is majorly seen in patients with MYD88L265P with ibrutinib monotherapy is dimensioned.  Patient's will respond well to rituximab plus ibrutinib irrespective to MYD88 mutation status.  Please comment on this.  https://www.nejm.org/doi/full/10.1056/NEJMoa1802917
  3. Please comment on MYD88 WT and association with disease transformation in WM (Zanwar, AJH, 2019). https://onlinelibrary.wiley.com/doi/abs/10.1002/ajh.25697
  4. Under section 7.2, please comment on using MASS spec to detect MRD status on patients with MM after ASCT. This is actually one of the good examples of clinical use of Mass-Spec to detect MRD status in patients and with this data, this section will be further enriched. https://ashpublications.org/blood/article/134/Supplement_1/4386/424666/Prognostic-Implications-of-Serum-Monoclonal
  5. Please add mantle cell lymphoma as well under other lymphoma hours with IgM monoclonal gammopathy.  This is a rare entity however it has been documented in the past.  https://pubmed.ncbi.nlm.nih.gov/15842043/
  6. Please comment in a separate section about IgM and coagulopathies such as cold hemolytic anemia (which this review has not addressed), cryoglobulinemia and cold agglutinin disease.  This are indeed a known entities associated with IgM paraproteinemia and should be included in this comprehensive review as a separate section.

Thank you

Author Response

please see the document attached

Reviewer 2 Report

Girard et al. gave a clinical overview of IgM paraproteinemia in this manuscript. It may be useful for physicians to evaluate patients with IgM gammopathies. However, the following several points should be addressed.

1) It is not necessary to present a clinical case in each category. Instead, the definition of the diagnosis should be clearly noted.

2) The clinical features caused by IgM gammopathies should be explained, i.e., neuropathy caused by antibodies to myelin-associated glycoprotein and other neural components.

3) Table 1 shows that the frequency of the MYD88 L265P mutation in IgM MGUS patients is 10%. However, some papers reported that the MYD88 L265P mutation was detected in approximately 40% of those patients. Thus, the authors noted the frequency correctly in the text. Furthermore, it would be better to note the frequency in previously treated WM patients.

4) IgM levels should be included and the frequency of MYD88 L265P was corrected in Table 1. Light Chain Amyloid=>Light chain amyloidosis.

5) Differential diagnosis between IgM MM and WM is very important, and thus the authors should describe more details of the diagnosis of IgM MM as well as note the prognosis of those patients.

6) Immunoparesis in WM patients is relatively mild compared with that in patients with other types of MM. This point should be noted and discussed.

7) It would be preferable to discuss why mature B-cell malignancy patients with IgM paraproteins have adverse outcomes.

Author Response

please see the document attached
